# Social–Emotional Skills Correlate with Reading Ability among Typically Developing Readers: A Meta-Analysis

**Liyan Yu [1]** , **Jane Jie Yu [2]** and **Xiuhong Tong [1],***

[1]   Department of Psychology, The Education University of Hong Kong, Hong Kong, China
[2]   Department of Sport and Exercise Science, College of Education, Zhejiang University,
     Hangzhou 310058, China
*   Correspondence: xhtong@eduhk.hk

**Abstract:** This meta-analysis examined the correlation between social–emotional skills and reading ability and identified possible moderators by synthesizing 285 correlations from 37 studies among 38 samples with 28,404 participants. Results showed a significantly positive correlation between social–emotional skills and reading ability among typically developing readers, $r = 0.23$, 95% CI [0.19, 0.28]. The moderation analysis revealed that, after controlling for types of social–emotional skills and grade level, the correlation between social–emotional skills and reading ability was moderated by the levels of reading (i.e., word reading vs. reading comprehension), $\beta = 0.07$, 95% CI [0.02, 0.11], $t = 5.03$, $p < 0.05$. Specifically, social–emotional skills had a significantly stronger correlation with reading comprehension than it with word reading. This work provides support for the lattice model of reading, suggesting that future research efforts are needed to examine the underlying mechanisms between social–emotional skills and reading ability.

**Keywords:** social–emotional skills; reading ability; meta-analysis

## 1. Introduction

Reading ability is crucial not only for academic success but also for participation in the workforce and society [1]. The lattice model of reading (LMR) proposes that the social–cognitive process plays an essential role in the development of reading ability (i.e., word reading and reading comprehension) [2]. The social–cognitive process includes executive function, motivation, metacognition, learning-related social skills, and emotional intelligence (e.g., self-motivation and emotion regulation) [2]. Previous studies examining the contribution of the social–cognitive process to reading mainly focused on the role of executive function and motivation [3,4], and less is known about the role of social skills and emotional intelligence. Social skills and emotional intelligence are defined as social–emotional skills in this study. Although an increasing number of studies examined the relationship between social–emotional skills and reading ability, the results are inconsistent, with some studies finding a significant correlation between them [5], while other paper did not find any [6]. It is also unclear whether the correlation between social–emotional skills and reading ability differs in various sub-groups. Understanding the correlation between emotional skills and reading ability and the potential moderators can provide information for future targeted policies and interventions to improve students' social–emotional skills and reading ability. To better understand the extent to which social–emotional skills (i.e., social skills and emotional intelligence) impact reading ability and the potential moderators, we conducted this meta-analysis—as far as we know, the first of its kind—that systematically tests the correlation between these two variables.

## 2. Social–Emotional Skills and Reading Ability

In this study, the social–emotional skills refer to social skills, i.e., the ability to function competently at social tasks [7], and emotional intelligence, i.e., the ability to understand

one's own and others' emotions, control one's emotions, and be self-motivated [8]. The reading ability refers to word reading, i.e., the ability to identify written words [9], and reading comprehension, the ability to understand the text [10]. According to LMR, social skills and emotional intelligence, which act as essential components in social–cognitive processes, play important roles in students' reading ability (i.e., word reading and reading comprehension) [2]. As social skills and emotional intelligence share certain components, e.g., assertiveness and interpersonal skills [11–15], both were considered in this meta-analysis.

LMR suggests that during early and middle childhood, social–emotional skills in the social–cognitive process have a reciprocal correlation with reading ability, each affecting the development of the other [2]. Indeed, some researchers argue that books, and in particular works of literature, are a resource of human emotions by which individuals can increase their emotional knowledge. In turn, understanding the nature, causes, and control/regulation of emotions expressed in reading materials can promote reading comprehension [16]. Prior studies support this theoretical hypothesis [17,18], suggesting that reading ability can also predict children's social–emotional skills. For example, Benner et al.'s longitudinal study [19] showed that from kindergarten to second grade, word reading (i.e., letter-word identification) and reading comprehension (i.e., passage comprehension) were significantly and positively related to children's social skills (i.e., cooperation, assertiveness, and self-control) [17]. Furthermore, word reading could account for children's social skills (i.e., social adjustment) [17], while reading difficulty might lead to social–emotional problems, such as antisocial emotional disorder [20]. Similarly, using the data from the Early Childhood Longitudinal Study program, Caemmerer and Keith [21] showed that social skills might have a reciprocal correlation with students' reading ability from kindergarten to the eighth grade. Another study among 468 first graders indicated that their reading comprehension in the fall could predict their social skills, indexed by cooperation, assertion, and self-control, in the winter, which in turn could predict their reading comprehension in the spring [22].

Several potential explanations may account for the correlation between social–emotional skills and reading ability. The first is that social–emotional skills may influence reading ability through cognitive ability. For example, emotions have been related to working memory [23], with readers of higher social–emotional skills, such as empathy, more likely to connect emotionally with characters in stories, a connection that helps them more accurately memorize the content of the reading material [23–25]. Social–emotional skills may also be related to nonverbal intelligence [26], as individuals with higher social–emotional skills are likely to be smarter than those with poor social–emotional skills [27]. Thus, readers with higher social–emotional skills are also more likely to be proficient readers. It is thereby highly possible that the relationship between social–emotional skills and reading ability might be mediated by cognitive abilities.

Second, effective social–emotional skills may help readers increase their language skills, such as vocabulary knowledge, which can then improve their understanding of reading materials. Children with high social–emotional skills may have more social interactions, during which vocabulary knowledge may develop through incidental learning. For example, one prior study showed that early mother–child conversations were positively related to children's vocabulary knowledge in kindergarten and second grade [28]. Another study indicated that Iranian English children's vocabulary learning could occur through computer-mediated communication and face-to-face interaction [29]. Vocabulary development facilitates the reading process since the core of reading is to extract meaning from written texts. Some theoretical models of reading even consider vocabulary knowledge as the foundation for proficient reading comprehension [30]. Vocabulary knowledge also contributes to the development of reading-related skills, such as the phonological process [31]. Thus, vocabulary knowledge could be a potential mediator in the relationship between social–emotional skills and reading ability.

Third, the emotioncy theory suggests that reading may be a sense- and emotion-related understanding process that is jointly influenced by our familiarity with the specific topic,

the frequency of exposure to that topic, and our emotions towards that topic [32]. In other words, our reading ability is correlated with our sensory experiences. We know that social interactions can be similar to situations in some reading materials. Individuals with higher social–emotional skills tend to acquire knowledge about diverse topics because they interact more with others, which means that they may be more familiar with the related topics and have a higher frequency of exposure to those topics. One previous study, for example, indicated that learning a new reading passage was easier for students with more sensory-related experiences [32].

Fourth, social–emotional skills may be related to reading ability through reading motivation. Since students with high social–emotional skills have more social interactions and communicate with others on many topics, they become more familiar with these topics. Children usually display higher reading motivation for familiar topics than for unfamiliar ones. For example, one prior study found that students with more sensory experiences (e.g., hearing more) on specific topics tended to have high reading motivation when reading material on these topics, which ultimately contributed to their reading comprehension [33]. Students with high social–emotional skills are also more likely to have good relationships with teachers and peers, which may influence students' school adjustment, including their motivation to learn and read. For instance, Li et al.'s study [34] found that peer relationships could influence Chinese adolescents' learning motivation at school. Thus, social–emotional skills may be correlated with reading ability via reading motivation.

Though the majority of past studies have reported that social–emotional skills are positive predictors of individuals' reading comprehension even after controlling for educational level and IQ [5,18,35], some studies did not find a significant correlation between social–emotional skills and reading ability. For instance, Pishghadam [36] found that emotional intelligence was not significantly related to undergraduate students' English reading comprehension. Similarly, a study among a sample of Australian fifth graders indicated that social skills were not associated with reading comprehension, $r = 0.08$ [37]. We examined this inconsistency by synchronizing past studies to systematically test the relationship between social–emotional skills and reading ability among different subgroups.

## 3. Potential Moderators

We explored whether three potential moderators: types of social–emotional skills, types of reading outcomes, and grade level, could explain the inconsistency in the correlation between social–emotional skills and reading ability.

### 3.1. Types of Social–Emotional Skills

Despite their overlap, social and emotional skills differ in some respects. From a social perspective, social skills refer to a range of verbal and nonverbal skills that individuals use during social interaction that can influence the responses of others without causing distress [38]. Theoretical models of emotions or emotional intelligence define emotional skills in different ways. For instance, Mayer and Salovey [39,40] proposed a four-branch hierarchical model of emotional skills indicating that emotional intelligence can be defined according to emotion-related abilities, namely, emotional perception, emotional facilitation of thought, emotional understanding, and emotional management. In sum, social skills focus on verbal and nonverbal skills needed during social tasks, such as communication and cooperation skills [7,14], while emotional intelligence focuses on emotion-related skills, such as stress management and empathy [11–13]. Thus, different aspects of social–emotional skills were assessed in past studies. The question of whether a specific type of social-emotional skill could moderate the relationship between social–emotional skills and reading ability naturally arose.

We postulated that some social–emotional skills might be more closely related to reading activities. For example, individuals with a high level of empathy are more likely to identify with characters and become deeply engaged in the reading material [24,25]. An engaged reader always has a high level of reading motivation [41,42], which is positively

and significantly related to individuals' reading ability [43]. Smith et al. [44] also found that American children's social skills (i.e., communication skills) at five years old were significantly correlated to reading comprehension at fourteen years old, $r = 0.32$, $p < 0.01$, but their emotional intelligence (i.e., frustration tolerance) at ten years old was not significantly correlated to reading comprehension, $r = 0.21$, $p > 0.05$. Therefore, the correlation between social–emotional skills might be moderated by a specific type of social–emotional skill. To verify this prediction, we evaluated the moderation effect of the types of social–emotional skills on the relationship between social–emotional skills and reading comprehension.

### 3.2. Types of Reading Outcomes

It is unclear whether the relationship between social–emotional skills and reading varies with different reading outcomes. In the present study, we included word reading and reading comprehension as reading outcomes. Word reading is the ability to recognize letters or words accurately and fluently [45,46]. Reading comprehension is the ability to read and understand a passage or text [10,47]. In a recent study, Sparapani and colleagues [22] tested the correlation between social–emotional skills and reading ability among first graders. The results showed that social–emotional skills were more strongly correlated with word reading ability (i.e., cooperation, $r = 0.47$; assertiveness, $r = 0.42$; and self-control, $r = 0.30$) than with reading comprehension ability (i.e., cooperation, $r = 0.36$; assertiveness, $r = 0.23$; and self-control, $r = 0.25$) [22]. This implies that social–emotional skills may play a more critical role in word-level reading in younger children. Thus, we tested whether the relationship between social–emotional skills and reading differs across different reading outcomes. We hypothesized that this relation could be moderated by the types of reading outcomes.

### 3.3. Grade Level

A prior meta-analysis found a significantly stronger association between emotional competence and academic achievement in primary school students than in college students [48], though it is unclear whether grade level moderated the relationship between social–emotional skills and reading ability. The correlation between social–emotional skills and reading ability may increase with educational level, with a stronger correlation for readers in high school or above than readers in preschool and primary school. One possibility is that readers with a higher educational level may have more emotional and social experiences to draw on and thus be able to better analyze the social relations among characters and their emotions in the reading material, which, in turn, contributes to their reading progress. Indeed, one prior longitudinal study showed that students' emotional vocabulary improved from the 10th to the 12th grade [49]. Similarly, another three-year longitudinal study found that children's social–emotional skills increased after they entered kindergarten [50].

However, MacCann et al. [51] suggested that younger children with poor emotional skills are more easily affected by psychological factors and unable to focus on their learning. Chall [52] suggested that children after nine years old tend to read to satisfy their desire to gain new knowledge, to learn new ideas and attitudes, and to experience new feelings and learn more efficiently than children below nine. Similarly, Boud [53] posited that older students become autonomous learners whose learning activities are less likely to be influenced by emotions. For example, correlational and experimental studies indicated that, after controlling for cognitive factors, social–emotional skills were significant predictors of reading ability in kindergarten students [17,54], primary school students [17], and secondary school students [54], with or without autism spectrum disorder. In contrast, another study showed that undergraduate students' social–emotional skills were not significantly related to their reading ability [55,56]. Therefore, grade level might moderate the relationship between social–emotional skills and reading abilities.

#### 4. The Present Study

This meta-analysis had two aims. First, we evaluated whether significant correlations exist between social–emotional skills and reading ability and, if so, what the strengths of these correlations are. Second, we examined whether the relationship between social–emotional skills and reading ability was moderated by the types of social–emotional skills (i.e., social skills and emotional intelligence), types of reading abilities (i.e., word reading and reading comprehension), and grade level. We predicted that: (1) social–emotional skills and reading ability would be significantly correlated with each other, and (2) the types of social–emotional skills, types of reading abilities, and grade levels would moderate the relationship between social–emotional skills and reading ability.

#### 5. Method

This meta-analysis was conducted and reported following the guidelines of the preferred reporting items for systematic reviews and Meta-analyses (PRISMA) [57]. This study was registered in the international prospective register of systematic reviews (PROSPERO) with reference number CRD 42020195841.

#### 6. Literature Search

*6.1. Electronic Database Search*

Five electronic databases (i.e., ProQuest, PsycARTICLES, PsycINFO, PubMed, and Web of Science) were used to identify relevant studies published in English before December 2021. The search terms were grouped into two categories: (1) reading ability (i.e., word reading and reading comprehension) and (2) social–emotional skills (i.e., social skills and emotional intelligence).

*6.2. Manual Search*

We also performed a manual search on Google Scholar based on these keywords: emotional intelligence, social skills, word reading, and reading comprehension. Six additional articles [36,56,58–61] were obtained through this manual search. Another paper [62] was identified by reviewing the references of the included papers.

#### 7. Inclusion and Exclusion Criteria

Articles were included in this meta-analysis if they met the following criteria: (1) they were papers published or unpublished in English; (2) they measured the relationship between social–emotional skills and reading ability; (3) the participants included in the studies were typically developing readers without any physical disabilities (e.g., hearing loss) or special education needs; (4) if the article reported the correlation among typically developing and nontypically developing children, respectively, only the correlation among the typically developing children was included; and (5) if the article reported the correlation between social–emotional skills and reading ability for a group in which the majority of participants were typically developing children, we included it. To illustrate (4), Schwab et al.'s study [37] reported the correlation between social–emotional skills and reading ability among students with and without special educational needs; we included only the data for students without special educational needs. To illustrate (5), we included Hall and DiPerna's study [63] since 92.10% of the participants were typically developing children.

Studies were excluded if: (1) they were books, (2) reviews or commentaries, or (3) qualitative or case studies; (4) all the participants were nontypically developing children, such as those with a learning disability [64,65], attention-deficit–hyperactivity disorder [66], or an autism spectrum disorder [67]; and (5) they reported the same correlation among the same group; in this case, we used only one study (e.g., of the two studies by Firdaus [60,68] reporting the correlation between emotional intelligence and reading comprehension for the same group, we included only one of them).

## 8. Paper Filtering

The paper filtering process is shown in Figure 1. The initial search yielded 5123 papers, of which 4376 were left after removing 747 duplications. Next, two reviewers (i.e., the first author and a research assistant (RA)) were trained to screen the title and abstract of the remaining records independently. During this step, we excluded: (1) qualitative or case studies; (2) meta-analyses or reviews; (3) neurophysiological studies; (4) papers that recruited only nontypically developing readers (e.g., those with a learning/reading disability, attention-deficit–hyperactivity disorder, an autism spectrum disorder, or other disabilities); and (5) papers that did not mention social–emotional skills or reading ability, of which there were 179. Finally, the same reviewers screened the full-text articles for inclusion. During this step, we excluded: (1) unrelated papers; (2) qualitative studies; (3) whole papers that were unavailable even after a request to the author's university library; and (4) papers that did not report the correlation between social–emotional skills and reading ability among typically developing readers (e.g., they reported the correlation only between social–emotional difficulty and reading ability or between social–emotional skills and reading difficulty). This filtering yielded 42 papers. Reviewing the references of all included papers yielded another paper. Thus, 43 papers with 44 samples met the inclusion criteria. As we identified six influential samples, 37 studies with 38 samples, including 285 effect sizes, were considered in our meta-analysis. The coding reliabilities for both title–abstract and full-text screening were greater than the excellent level of agreement (k = 0.86 and 0.94, respectively, which is >0.75) [69]. Any disagreements arising in the aforementioned phases were resolved via discussion with the other two authors. The final agreement was 100%.

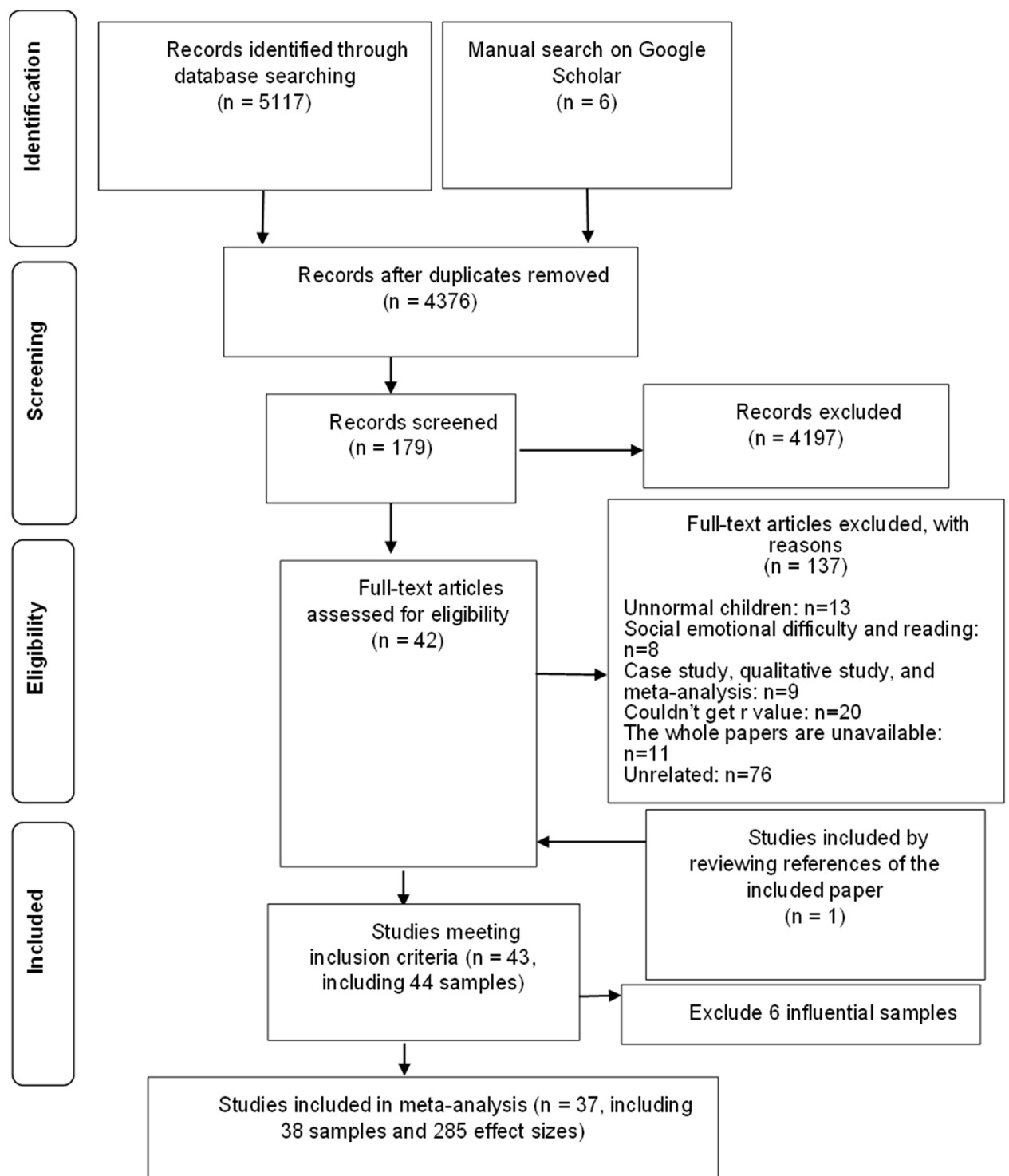

**Figure 1.** Flow diagram for the search and inclusion criteria for studies in the present review.

## 9. Data Coding

The two reviewers independently coded the included studies. All discrepancies were resolved through discussion with the other two authors before data analysis. The percentage of agreement for all coded variables was 100%. All studies that met the inclusion criteria were extracted based on the following aspects: the design (i.e., cross-sectional and longitudinal), grade level (i.e., kindergarten, primary school, secondary school, and university or above), sample size, gender (the percentage of males among the participants), studied language, types of social–emotional skills (i.e., emotional intelligence and social

skills), types of reading ability (i.e., word reading and reading comprehension), and Pearson's correlation (*r*) between social–emotional skills and reading ability (for details, see Table 1). The sample size (*n*) indicated the number of participants who completed the social–emotional skills scale and reading ability test. If the study did not report the number of participants who completed the social–emotional skills scale and reading ability test, the number of participants who took part in the study was coded as the sample size. If a study reported more than one *r* value between social–emotional skills and reading ability for one sample, all the *r* values were included in the present study.

**Table 1.** Summary of key features coded for studies meeting inclusion criteria.

| Author, Year of Publication | Design [a] | Grade [b] | N [c] | ES [d] | Gender (Male Percentage) | Language | Outcome Measures | | r | Quality of Methodology | | |
| | | | | | | | Social–Emotional Skills [e] | Reading [f] | | Sample [h] | Measurement [h] | Analysis [h] |
|---|---|---|---|---|---|---|---|---|---|---|---|---|
| Abdolrezapour and Tavakoli [59] | C | Null | 63 | 1 | Null | English | EI | RC | 0.66 | * | ** | ** |
| Abdorazik [70] | C | U | 49 | 1 | 32.35% | English | EI | RC | 0.25 | ** | ** | *** |
| Agostin and Bain [71] | L | K | 184 | 3 | 47.30% | English | SS | WR | 0.08–0.29 | ** | ** | ** |
| Alipanahi and Tariverdi [72] | C | U | 50 | 1 | 60.00% | English | EI | RC | 0.68 | ** | *** | ** |
| Arabsarhangi and Noroozi [5] | C | Null | 50 | 1 | Null | English | EI | RC | 0.70 | ** | ** | *** |
| Arnold et al. [73] | C | K | 467 | 1 | 51.82% | English | SS | WR | 0.13 | ** | *** | ** |
| Aryanto [74] | C | U | 46 | 1 | Null | English | EI | RC | 0.16 | * | ** | ** |
| Ates [75] | C | S | 138 | 1 | 29.71% | Turkish | EI | RC | 0.35 | ** | *** | ** |
| Benner et al. [17] | C | K and P | 150 | 1 | 54.00% | English | SS | WR | 0.49 | ** | ** | ** |
| Caemmerer and Keith [21] | L | K | 7802 | 120 | 49.40% | English | SS | RC | 0.17–0.31 | ** | *** | ** |
| Chen and Zhang [76] | C | U | 72 | 3 | 66.67% | English | EI | RC | 0.03–0.14 | ** | *** | ** |
| Farajnezhad and Tabatabai [77] | C | U | 150 | 1 | 63.33% | English | EI | RC | 0.18 | * | ** | ** |
| Feshbach and Feshbach [78] | L | P | 133 | 2 | 50.00% | English | EI | Null | −0.08–0.11 | ** | *** | ** |
| Firdaus [68] | C | S | 53 | 1 | Null | English | EI | RC | 0.66 | * | *** | ** |
| Froiland and Davison [79] | C | Null | 40 | 1 | 45.00% | English | EI | RC | 0.45 | ** | ** | ** |
| Ghabanchi and Rastegarl [6] | C | U | 55 | 1 | 45.45% | English | EI | RC | 0.19 | * | ** | ** |
| Hall and DiPerna [63] | L | P | 9225 | 4 | 50.20% | English | SS | RC | 0.10–0.14 | ** | *** | ** |
| Hindman and Morrison [80] | C | K | 229 | 2 | 50.00% | English | SS | WR | 0.10–0.17 | ** | *** | ** |
| Karbalaei and Sanati [81] | C | Null | 65 | 1 | Null | English | EI | RC | 0.84 | ** | ** | ** |
| Konold et al. [82] | L | K | 1102 | 96 | 51.00% | English | SS | WR | 0.01–0.31 | ** | *** | ** |
| Medford and McGeown [83] | L | K | 85 | 2 | 49.41% | English | SS | WR | 0.15–0.20 | ** | *** | ** |
| Montroy et al. [84] | L | K | 103 | 2 | 66.10% | English | SS | WR | 0.29–0.32 | ** | *** | *** |
| Motallebzadeh [62] | C | U | 170 | 1 | 22.80% | English | EI | RC | 0.54 | ** | ** | ** |
| Ortiz et al. [85] | L | K | 203 | 3 | 43.75% | English | SS | RC and WR | 0.24–0.25 | ** | *** | ** |
| Martí et al. [86] | C | P | 180 | 1 | 52.80% | Null | EI | RC | 0.21 | *** | ** | ** |
| Pishghadam and Tatataba'ian [87] | C | U | 90 | 1 | 44.44% | English | EI | RC | 0.17 | ** | *** | ** |
| Pishghadam [36] | C | U | 508 | 1 | 26.38% | English | EI | RC | 0.06 | ** | ** | ** |
| Ponitz et al. [88] | L | K | 343 | 1 | 48.00% | English | SS | WR | 0.15 | ** | *** | ** |
| Quirk et al. [89] | L | K | 112 | 2 | 55.00% | English | SE and EI | WR | 0.00–0.37 | ** | *** | ** |
| Razza et al. [90] | L | K | 669 | 2 | 47.83% | English | SS | RC | 0.25–0.27 | ** | *** | ** |
| Saeidi and Yusefi [91] | C | U | 143 | 1 | 18.88% | English | EI | RC | 0.20 | ** | ** | ** |
| Sartika et al. [92] | C | S | 140 | 1 | Null | Null | EI | RC | 0.24 | * | ** | ** |
| Schwab et al. [37] | L | P | 18 | 12 | 27.78% | German | SS and EI | RC and WR | −0.26–0.31 | ** | *** | ** |
| Smith et al. [44] | L | K | 79 | 4 | 52.00% | English | SS and EI | RC and WR | 0.21–0.34 | ** | ** | *** |
| Smith-Adcock et al. [93] | L | K | 3444 | 3 | 48.30% | English | SS | RC | 0.20–0.42 | ** | *** | ** |
| Sparapani et al. [22] | L | P | 449 | 3 | 46.00% | English | SS | RC and WR | 0.32–0.37 | ** | *** | ** |

**Table 1.** *Cont.*

| Author, Year of Publication | Design [a] | Grade [b] | N [c] | ES [d] | Gender (Male Percentage) | Language | Outcome Measures | | r | Quality of Methodology | | |
|---|---|---|---|---|---|---|---|---|---|---|---|---|
| | | | | | | | Social–Emotional Skills [e] | Reading [f] | | Sample [h] | Measurement [h] | Analysis [h] |
| Tabrizi and Esmaeili [94] | C | S | 121 | 1 | 0.00% | English | EI | RC | 0.55 | ** | ** | ** |
| Talebinejad and Fard [95] | C | S | 80 | 1 | 0.00% | English | EI | RC | 0.80 | ** | ** | ** |
| Wang and Algozzine [96] | C | P | 257 | 2 | 57.98% | English | SS | RC and WR | 0.28–0.32 | ** | *** | ** |
| Yu and Tong, 2023 [97] [g] | C | P | 679 | 1 | 50.70% | Chinese | EI | RC | 0.20 | *** | *** | *** |
| Zandi [55] | C | U | 239 | 1 | 0.00% | French | EI | RC | 0.12 | * | ** | ** |
| Zandi [61] | C | U | 200 | 1 | 100.00% | French | EI | RC | 0.15 | * | ** | ** |
| Zarezadeh [56] | C | U | 330 | 1 | Null | English | EI | RC | 0.18 | ** | ** | ** |

Notes: This table presents key study features only. Null means did not report. [a] Design was coded as C (cross-sectional) and L(longitudinal). [b] Grade was coded as K (kindergarten), P (primary school), S (secondary school), and U (university or above). [c] N = Sample size is reported as the maximum number of participants for calculating correlations. [d] ES = number of effect sizes (*r*) provided in the study. [e] Social–emotional skills were coded as EI (emotional intelligence), SS (social skills), and SE (social–emotional skills mixed). [f] Reading was coded as WR (word reading) and RC (reading comprehension). [g] This paper was unpublished. [h] Each criterion was assigned one star if the study did not meet any criteria, two stars if the study partially met the criteria or if the report was unclear, and three stars if the study fully met the criteria. * $p < 0.05$, ** $p < 0.01$, *** $p < 0.001$

## 10. Quality Assessment

The first author and the RA independently evaluated the quality of methodology for each article using three aspects of the McMaster Critical Review Form (i.e., sample, measurement, and analyses) [98,99]. For the sample criterion, studies were evaluated based on the reduction of selection bias, sample size, and details related to participants' characteristics. Sample size indicates whether the sample was large enough to conduct that study, and a sample size greater than 10 cases per parameter meets this criterion [100]. For the measurement criterion, papers were assessed based on the validity and reliability of both the social–emotional skills scale and the reading ability task. For the analysis criterion, statistical significance, provision of point estimates and variability, and the discussion of practical importance were considered. Each criterion was assigned one star if the study did not meet any criteria, two stars if the study partially met the criteria or if the report was unclear, and three stars if the study fully met the criteria [98].

The interrater reliability between the two researchers was 100% from all three aspects: sample, measurement, and analyses. Table 1 summarizes the characteristics of the included studies and presents ratings on quality assessment. Results showed that for the Sample criterion, eight studies used a representative sample, one study calculated the power of sample size, and eight studies did not provide the participants' grade level or gender. For the Measurement criterion, the social–emotional skills scale in all 43 studies was successfully used in previous studies, four studies did not report the task used to measure participants' reading ability, and 22 studies reported the reliability of both the social–emotional skills scale and reading ability task. For the Analysis criterion, four studies provided the confidence interval, and 40 studies reported the significance of the results and thoroughly discussed the implications of the findings.

## 11. Statistical Analysis Strategy

To conduct this meta-analysis, three packages (i.e., METACOR [101], METAFOR [102], and ROBUMETA [103]) in R were used (see Figure 2). Pearson's correlation (*r*) between social–emotional skills and reading ability was transferred to Fisher's *z*, and Fisher's *z* was used to indicate effect size in the meta-analysis. When examining the influential samples and publication bias, we calculated the mean for each sample with more than one effect size before calculating the DFBETAS and COVRATIO values and the *z* value. After excluding the effect sizes of all influential samples, to examine the overall correlation between social–emotional skills and reading ability and potential moderators, we handled the samples with more than one effect size using robust variance estimation (RVE) [104].

a
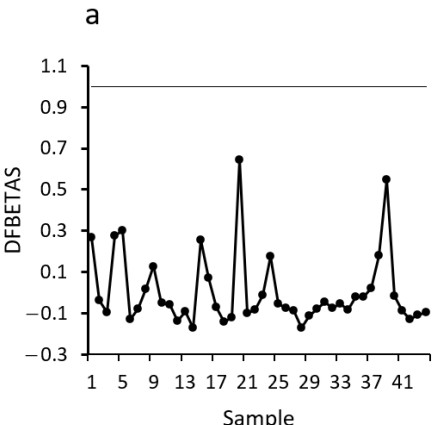

b
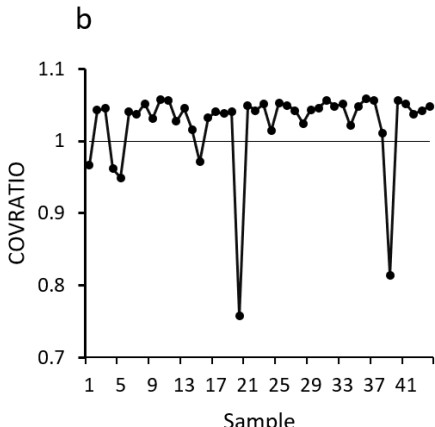

**Figure 2.** Plot of the (**a**) DFBETAS values and (**b**) COVRATIO values for 44 samples examining the correlation between social–emotional skills and reading ability. Note: for the samples with more than one effect size, the mean value was used to calculate each sample's DFBETAS and COVRATIO values.

Specifically, Pearson's *r* was first converted to Fisher's *z* using the DerSimonian–Laird (DSL) random-effect meta-analytical approach in the "metacor.DSL" function of the META-COR package. Second, we excluded the effect sizes for samples with an extremely high or lower average correlation between social–emotional skills and reading comprehension. In this step, the mean value was used for each sample with more than one effect size. If a study reported the correlation between social–emotional skills and reading ability for two or more independent samples, each sample was treated separately when conducting the meta-analysis. For example, Caemmerer and Keith's study [21] reported the correlation between social skills and reading ability of both the validation and the calibration samples; our meta-analysis coded this study to two independent samples. Then, we identified the influential samples by calculating the DFBETAS value and the COVRATIO value of each sample using the "influence" function in the METAFOR package. The DFBETAS value refers to the effect on the estimates for the regression coefficients when excluding each study [105], and the COVRATIO value refers to the effect on the variance–covariance matrix of the estimates when excluding each study [105]. A sample with a DFBETAS value greater than 1 [106] or a COVRATIO value smaller than 1 [107] could be identified as an influential sample and would be excluded in the later meta-analysis to generate a more precise estimation. We excluded all the effect sizes of six influential samples before calculating the overall correlation and the moderator analysis. Therefore, 37 studies with 38 samples, including 285 effect sizes, were considered in the fourth step.

Third, we examined the publication bias by using the "rma" function, the "regtest" function, and the "funnel plot" function in the METAFOR package. The funnel plot shows the distribution of the effect size of each sample, and an even distribution on the left and right sides of the mean effect size suggests no publication bias. The "regtest" function indicates the significance of publication bias with a mixed-effects meta-regression model. In this model, the standard error of each sample's effect size acted as the predictor, and each sample's effect size acted as the dependent variable [108,109].

Fourth, we calculated the overall correlation between social–emotional skills and reading ability and conducted subgroup analysis and moderator analysis using the "robu" function in the ROBUMETA package with 37 studies with 38 samples, including 285 effect sizes. Robust variance estimation (RVE) was used to deal with dependent effect sizes [104]. For the moderator analysis, three moderators (i.e., social–emotional skills, reading ability, and grade level) were entered into one model when performing the meta-regression. The *I*-squared, which refers to the percentage of variation among the studies resulting from heterogeneity rather than chance [110,111], was calculated to check heterogeneity.

## 12. Results

There were 5123 studies located in the initial search. After removing 747 duplicates, 4376 articles were screened by their titles and abstracts, with 179 further screened by full texts using the preestablished criteria (see Figure 1). Eventually, 43 studies among 28,765 participants met the inclusion criteria, and 37 studies among 38 samples with 28,404 participants were included in the meta-analysis after excluding six influential samples (see Table 1 for details of all papers meeting inclusion criteria).

## 13. Study Characteristics

As shown in Table 1, of the 43 studies, 28 were cross-sectional (65.12%), and 15 were longitudinal (34.88%). Twenty-five studies (58.14%) measured participants' emotional intelligence, 15 (34.88%) measured social skills, two (4.65%) measured emotional intelligence and social skills, and one (2.33%) measured emotional skills and social–emotional skills. Twenty-eight studies (65.12%) measured participants' reading comprehension, nine (20.93%) measured participants' word reading, five (11.63%) measured participants' word reading and reading comprehension, and one (2.33%) did not report the type of reading ability. Thirty-six studies (83.72%) measured participants' reading ability in English, two (4.65%) in French, one (2.33%) in German, one (2.33%) in Turkish, one (2.33%) in Chinese, and two (4.65%) did not report the language measured. All studies reported Pearson's *r* value between social–emotional skills and reading ability. The *r* value ranged from −0.26 to 0.84.

## 14. Participant Characteristics

Participants' sample sizes ranged from 18 to 9225, and grade levels ranged from kindergarten to post-undergraduate. In terms of gender, thirty-two studies (74.42%) included male and female participants, with the male percentage ranging from 18.88% to 66.67%, three studies (6.98%) included only female participants, one study (2.33%) included only male participants, and seven studies (16.28%) did not report participants' gender. The participants were recruited from kindergarten in 13 studies (30.23%), primary school in seven studies (16.28%), secondary school in five studies (11.63%), university or above in 13 studies (30.23%), both kindergarten and primary school in one study (2.33%), and four studies (9.30%) did not report participants' grade level.

## 15. Influential Samples

Each study's DFBETAS and COVRATIO values were calculated to determine whether the study was an influential sample. When analyzing small to medium data sets, a study with a DFBETAS value greater than one is often considered an influential sample [106,107]. As shown in Figure 2, the DFBETAS values of all 44 samples were smaller than 1. No influential samples were found based on this value.

When a study's COVRATIO value is smaller than 1, the removal of that study can yield a more precise estimation [107]. As shown in Figure 2, the COVRATIO value for Sample 1 [59], Sample 4 [72], Sample 5 [5], Sample 15 [68], Sample 20 [81], and Sample 39 [95] were below 1. These studies were identified as influential samples and removed in the later analysis.

## 16. Publication Bias

Publication bias means that journals prefer studies with statistically significant results [112]. The funnel plot of the distribution of effect sizes and Egger's regression test were conducted to examine publication bias. As shown in Figure 3, the effect sizes of the studies were, to some extent, symmetrically distributed around the overall effect size, suggesting no publication bias. Many papers did not plot in the white-shaded area of the triangle, suggesting high heterogeneity among the studies. Egger's regression test showed that the standard error of the effect size could not predict the effect size, with $z = 0.50$ and

$p = 0.62$, indicating that the statistical significance of the paper's results would not influence its publication.

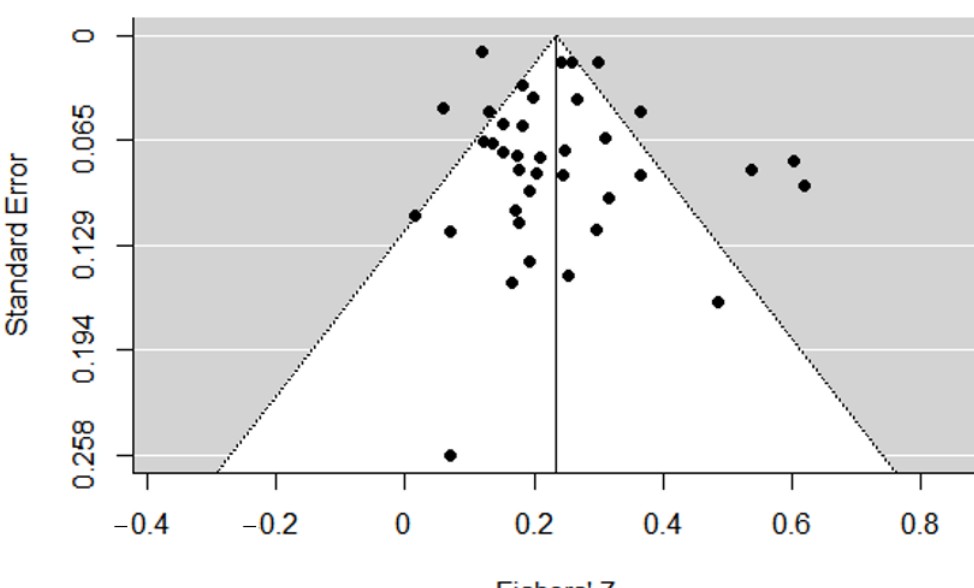

**Figure 3.** Funnel plot for the overall analysis of the social–emotional skills with reading ability after excluding influential samples. Note: for the samples with more than one effect size, the mean value was used to calculate publication bias.

## 17. Overall Correlation between Social–Emotional Skills and Reading Ability

The "robu" function in the ROBUMETA package was used to calculate the overall effect size between social–emotional skills and reading ability. The result indicated that social–emotional skills were moderately and significantly correlated with reading ability, $r = 0.23$, 95% CI [0.19, 0.28], $p < 0.001$ (see Table 2 and Figure 4).

**Table 2.** Relationship between social–emotional skills and reading ability after excluding the influential samples.

| Measures | NS [a] | ES [b] | Fisher's Z | 95% CI | Tau. sq [c] | *p*-Value |
|---|---|---|---|---|---|---|
| Overall | 38 | 285 | 0.23 | [0.19, 0.28] | 0.011 | <0.001 *** |
| Social–emotional skills [d] | | | | | | |
| Emotional intelligence | 22 | 30 | 0.22 | [0.14, 0.30] | 0.017 | <0.001 *** |
| Social skills | 19 | 254 | 0.24 | [0.19, 0.29] | 0.010 | <0.001 *** |
| Reading ability [e] | | | | | | |
| Word reading | 14 | 120 | 0.25 | [0.17, 0.32] | 0.012 | <0.001 *** |
| Reading comprehension | 28 | 163 | 0.25 | [0.20, 0.30] | 0.011 | <0.001 *** |
| Grade level [f] | | | | | | |
| Kindergarten | 14 | 89 | 0.22 | [0.18, 0.27] | 0.006 | <0.001 *** |
| Primary school | 13 | 155 | 0.22 | [0.16, 0.27] | 0.007 | <0.001 *** |
| Secondary school | 6 | 25 | 0.29 | [0.13, 0.44] | 0.011 | 0.006 ** |
| University and above | 12 | 14 | 0.20 | [0.09, 0.30] | 0.016 | 0.001 ** |

Notes: [a] NS = number of samples; [b] ES = number of effect sizes; [c] Tau. sq = between-study sampling variance. [d] Studies mixing the emotional intelligence and social skills together were not reported here, but were considered when calculating the overall correlation. [e] Studies mixing the word reading and reading comprehension together were not reported here, but were considered when calculating the overall correlation. [f] Studies did not report the grade of students or mixed above four categories together were not shown here, but were included when calculating the overall correlation. Robust variance estimation (RVE) was used to handle the samples with more than one effect size after excluding the effect sizes of all influential samples. ** $p < 0.01$, *** $p < 0.001$.

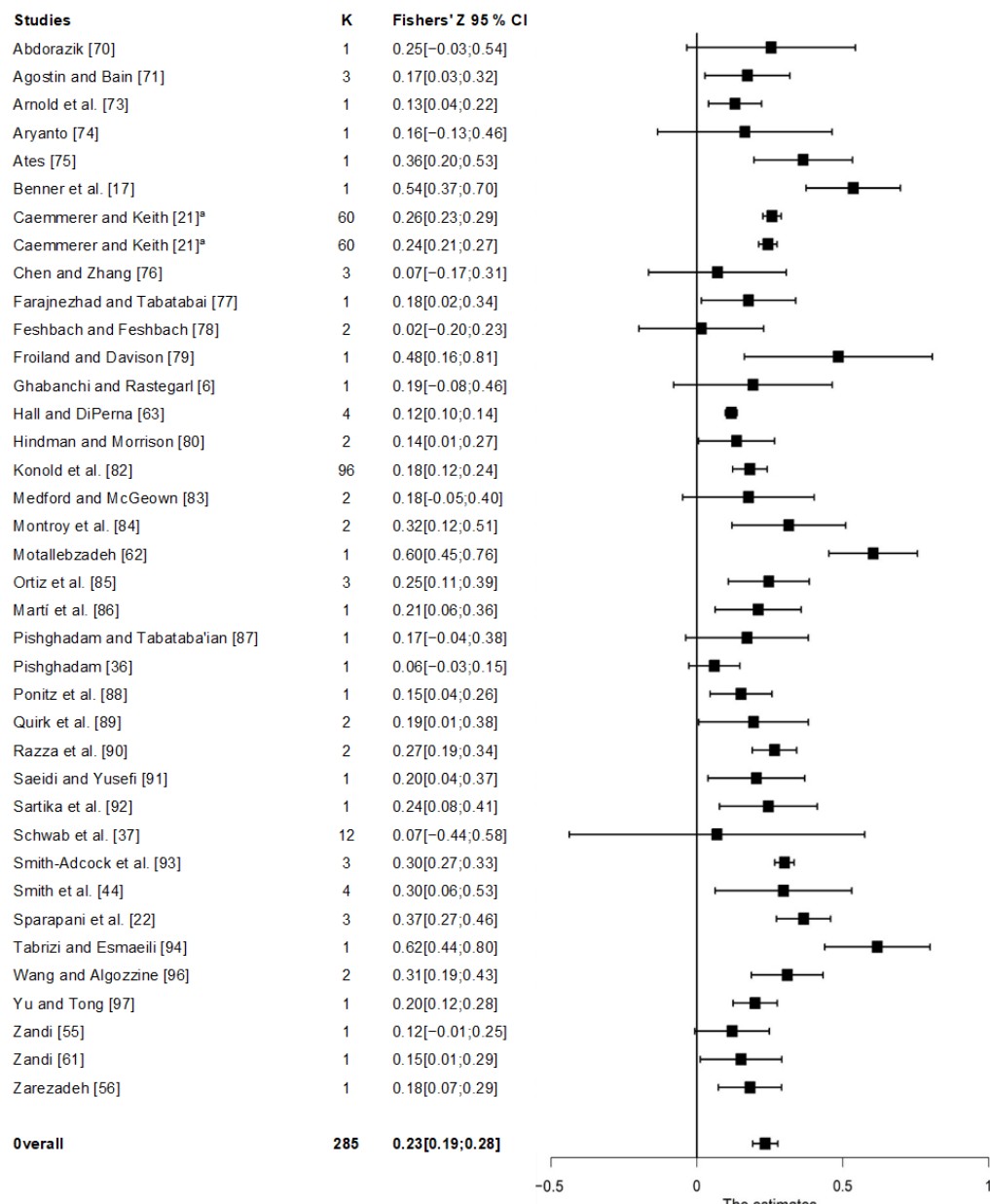

**Figure 4.** Overall correlation between the social–emotional skills and reading ability after excluding effect sizes of influential samples. Note: K refers to the number of effect size for each sample. [a] This study reported the r values for two independent samples.

## 18. Moderation Effects

*18.1. Moderation Effect of the Types of Social–Emotional Skills*

In this study, we coded social–emotional skills as emotional intelligence (30 effect sizes) and social skills (254 effect sizes). As shown in Table 2, reading ability was significantly correlated with emotional intelligence, $r = 0.22$, 95% CI [0.14, 0.30], $p < 0.001$, and social skills, $r = 0.24$, 95% CI [0.19, 0.29], $p < 0.001$. As shown in Table 3, after controlling for types of reading ability and grade level, emotional intelligence and social skills were related to a similar degree to reading ability, $\beta = 0.00$, 95% CI [−0.11, 0.10], $t = −0.09$, $p = 0.927$, which suggested that the relationship between social–emotional skills and reading ability was not affected by the types of social–emotional skills.

**Table 3.** Meta-regression of the moderation analysis on the relationship between social–emotional skills and reading ability.

| Measure | Beta | SE | t | 95% CI | p |
|---|---|---|---|---|---|
| Social–emotional skills | | | | | |
| Emotional intelligence vs. social skills | 0.00 | 0.05 | −0.09 | [−0.11, 0.10] | 0.927 |
| Reading ability | | | | | |
| Word reading vs. reading comprehension | 0.07 | 0.01 | 5.03 | [0.02, 0.11] | 0.022 * |
| Grade level | | | | | |
| Kindergarten vs. primary school | 0.02 | 0.01 | 2.90 | [−0.004, 0.05] | 0.070 [†] |
| Kindergarten vs. secondary school | 0.02 | 0.02 | 1.07 | [−0.07, 0.10] | 0.410 |
| Kindergarten vs. university | −0.06 | 0.06 | −0.98 | [−0.19, 0.07] | 0.344 |
| Primary school vs. secondary school | −0.005 | 0.01 | −0.39 | [−0.07, 0.06] | 0.741 |
| Primary school vs. university | −0.08 | 0.06 | −1.37 | [−0.22, 0.05] | 0.194 |
| Secondary school vs. university | −0.08 | 0.06 | −1.24 | [−0.22, 0.06] | 0.233 |

Notes: All moderators were entered in one model. Several models were run for the variable with more than two categories (i.e., grade level). For the convenience of presentation, subgroup comparisons within categorical moderators are all listed in the table. The first group in each group comparison variable is the reference group (e.g., in word reading versus reading comprehension, word reading is the reference group in the dummy coding of reading ability). Number of samples for this model is 38. Number of correlations for this model is 285. Between-study sampling variance ($\tau^2$) for this model is 0.001. Robust variance estimation (RVE) was used to handle the samples with more than one effect size after excluding the effect sizes of all influential samples. [†] $p < 0.1$, * $p < 0.05$.

*18.2. Moderation Effect of Types of Reading Ability*

The outcomes of reading ability were categorized into two types (i.e., word reading, including 120 effect sizes; and reading comprehension, including 163 effect sizes). As shown in Table 2, the average correlations between social-emotional skills and both types of reading ability were significant ($r = 0.25$, 95% CI [0.17, 0.32], $p < 0.001$ for word reading, and $r = 0.25$, 95% CI [0.20, 0.30], $p < 0.001$ for reading comprehension). Additionally, after controlling for types of social–emotional skills and grade level, reading comprehension was more strongly related to social–emotional skills than word reading, β = 0.07, 95% CI [0.02, 0.11], $t = 5.03$, $p = 0.022$ (see Table 3), suggesting that the relationship between social—emotional skills and reading ability did vary with types of reading ability.

*18.3. Moderation Effect of Grade Level*

In this study, we coded grade level into four groups (i.e., kindergarten, including 89 effect sizes; primary school, including 155 effect sizes; secondary school, including 25 effect sizes; and university or above, including 14 effect sizes). As shown in Table 2, the average relationship between social–emotional skills and reading ability was significant for all four groups: kindergarten, $r = 0.22$, 95% CI [0.18, 0.27], $p < 0.001$; primary school, $r = 0.22$, 95% CI [0.16, 0.27], $p < 0.001$; secondary school, $r = 0.29$, 95% CI [0.13, 0.44], $p = 0.006$; and university or above, $r = 0.20$, 95% CI [0.09, 0.30], $p = 0.001$. As shown in Table 3, after controlling for types of social–emotional skills and types of reading ability, a numerical tendency for a bigger magnitude in the correlation between social–emotional skills and reading ability occurred for elementary school students than for kindergarteners, β = 0.02, 95% CI [−0.004, 0.05], $t = 2.90$, $p = 0.070$. Given the small sample size, the correlation between social–emotional skills and reading ability may vary across grade levels.

## 19. Discussion

The present study examined the overall correlation between social–emotional skills and reading ability and the potential moderators of this correlation. Consistent with LMR, this study showed that the overall correlation between social–emotional skills and reading ability was significant, $r = 0.23$, $p < 0.001$. In addition, the moderator analysis indicated that types of reading ability could moderate the correlation between social–emotional skills and reading ability after controlling for types of social–emotional skills and grade level, with reading comprehension showing a stronger correlation with social–emotional skills

than with word reading. Contrary to our hypothesis, the moderating effect of types of social-emotional skills and grade level were not significant.

## 20. The Relationship between Social–Emotional Skills and Reading Ability

This meta-analysis is the first to test the correlation between social–emotional skills and reading ability guided with the LMR [2]. Consistent with the LMR, this study found a significant and moderate correlation between social–emotional skills and reading ability. This finding also aligns with many previous studies, indicating a significantly positive correlation between social–emotional skills and reading ability. For example, in a sample of American adolescents aged 16 to 18, Froiland and Davison [79] found that social–emotional skills (i.e., social perception) could predict students' reading comprehension even after controlling gender, grade, and ethnicity. In a longitudinal study, Ortiz et al. [85] found that children's social–emotional skills (i.e., social skills) in kindergarten were significantly and positively related to their word reading in kindergarten and reading comprehension at grade one. Similarly, in a sample of English learners in grade eleven, Firdaus [60] found that students' social–emotional skills (i.e., emotional skills) could predict their reading comprehension.

Several potential explanations exist for the correlation between social–emotional skills and reading ability. First, students with high social–emotional skills are more likely to build and maintain friendly relationships with teachers [113] and peers [114], which enables them to learn efficiently at school [115], and ultimately result in good academic performance including reading. For example, in a sample of college students, Schutte et al. [116] found that students' social–emotional skills (i.e., emotional skills) were positively related to their scores in close and affectionate relationships. Another previous study also showed that college students' social–emotional skills (i.e., emotional management) could predict positive interpersonal relationships, and students with higher social–emotional skills tended to report fewer negative interactions with close friends [117]. The interpersonal relationship was related to children's academic motivation at school. A prior study conducted on a sample of Chinese adolescents aged 13 to 17 indicated that peer relationships could influence their academic motivation, which could affect academic performance (i.e., mathematics achievement) [34]. Future studies should further examine whether social–emotional skills influence reading ability through the interpersonal relationship at school.

Second, individuals with higher social–emotional skills may have more high-quality social interactions and communications, causing their verbal skills to improve through incidental learning, which is ultimately related to reading ability [118]. In a sample of Spanish preschoolers who learned English as a second language, Winsler et al. [119] found that children's social–emotional skills (i.e., initiative, self-control, and attachment with adults) at age four could predict their oral language skills in English at the end of kindergarten. Similarly, some other studies reported that social–emotional skills were positively related to specific language skills, such as vocabulary knowledge [28] and listening comprehension [79]. Language skills, such as vocabulary knowledge and listening comprehension, play essential roles in reading development [2]. Thus, social-emotional skills may be related to reading ability through language skills.

Additionally, there are some similar scenarios in reading comprehension and social interactions. Reading comprehension, especially when reading fiction, is a transportation into a narrative world that involves cognitive, emotional, and creative–imaginary processes [42]. Exposed to characters in different roles and various scenarios similar to social interactions in life, young readers may draw on their prior experiences to understand the emotions and feelings of each character, which enables them to practice and forge their social–emotional skills [120]. Vice versa, the frequency of exposure to a specific scenario and familiarity with this scenario will influence children's understanding of reading materials related to this scenario [32]. Therefore, children who struggle to draw inferences and conclusions when reading a text might also have difficulty understanding others' perspectives and predicting

others' intentions during social interactions, and children with lower social–emotional skills perform worse in reading tasks [121,122].

## 21. Moderation Effect of Types of Reading Outcomes

We also found that the types of social–emotional skills significantly moderated the relationship between social–emotional skills and reading comprehension, but this moderating effect was not significant for the correlation between social–emotional skills and word reading. This finding is quite understandable. First, reading comprehension is relatively more complicated than word reading and involves both lower-level word processing skills and higher-level cognitive–linguistic processing skills [123]. Readers with higher social–emotional skills "would be able to carry out higher-level processing more effectively and efficiently" [59]. In contrast, word reading can become automatic with increasing learning, which may not be much associated with individuals' social–emotional skills [124]. The CMR model posits that social–emotional skills, which are important elements of the psychological domain, could significantly and directly relate to reading comprehension [17] but may indirectly relate to word reading through reading comprehension [2].

## 22. Moderation Effect of Types of Social–Emotional Skills

Inconsistent with our prediction, types of social–emotional skills did not moderate the magnitude of the correlation between social–emotional skills and reading comprehension. One potential explanation for this unexpected result is the highly overlapped items used to measure social skills and emotional intelligence. Interpersonal skills, for instance, were used to measure social skills and emotional intelligence in the studies reviewed. Specifically, 16 of the 28 studies measured participants' emotional intelligence using the Bar-On emotional quotient inventory [11–13] (N = 13) or the trait emotional intelligence questionnaire [125] (N = 3), both of which include interpersonal skills as an index of emotional intelligence. Six studies considered interpersonal skills as components of social skills measured by the social skills rating system [15]. Therefore, the nonsignificant moderating effect of types of social–emotional skills may be due to overlaps between social skills and emotional intelligence.

## 23. Moderation Effect of Grade Level

Inconsistent with our prediction, the grade level could not influence the correlation between social–emotional skills and reading ability. This result is inconsistent with one prior study showing that the correlation between emotional intelligence and academic performance was more robust for lower-grade students than for higher-grade students [48]. This result is consistent with another prior study, indicating that the correlation between emotional intelligence and academic performance was not influenced by age [51]. The insignificant moderating effect of the grade level should be interpreted cautiously. Although we did not find significant differences in the correlation between social–emotional skills and reading ability across grade levels, the correlational value of secondary school students was numerically larger than that of primary school students (i.e., 0.29 versus 0.22). The nonsignificant moderating effect of grade level may be due to the limited studies conducted in secondary schools. As shown in Table 2, only six studies recruited participants from secondary schools. Future studies with bigger sample sizes should be conducted further to examine the moderating effect of the grade level

## 24. Limitations

This study has at least three limitations. First, we focused on typically developing individuals only, which may limit the generalization of the findings to clinical samples. For example, previous studies also showed that social–emotional skills were significantly correlated with the reading ability of students with autism spectrum disorders [126,127]. Future studies should examine whether the results can be extended to disabled groups and whether the correlation between social–emotional skills and reading ability varies in normal and disabled groups. Second, at 38 samples, after excluding influential samples,

our sample size for the overall meta-analysis was sufficient, but the sample sizes of some subgroups were insufficient. For example, the subgroup of secondary schools had only six samples, slightly larger than four, which, according to Fu et al. [128], is the minimum sample for a categorical subgroup variable. This low sample size may account for our inability to find a significant moderator effect of grade level, although the correlation for secondary schools ($r$ = 0.29) was statistically higher than other grade levels ($r$s = 0.20–0.29). Finally, we did not examine the moderation effect of IQ in the present study. Though we initially considered IQ as a potential moderator, we found that most studies did not report IQ scores, resulting in many missing values. The few studies that did measure participants' IQ used different tasks with different scales and provided the raw score of IQ. Therefore, including IQ with different scales as a moderator was statistically infeasible. Instead, the moderators used in this study were grade level, a categorical variable divided by grade; types of social–emotional skills, a categorical variable divided into social skills and emotional intelligence; and reading ability, a categorical variable divided into word reading and reading comprehension.

## 25. Implications

This study was the first to systematically examine the correlation between social–emotional skills and reading ability. The findings have implications for theories of reading as well as for instructions to improve students' social–emotional skills and reading ability. Specifically, we examined the correlation between social–emotional skills and reading ability based on LMR. In support of LMR, we found a significant correlation between social–emotional skills and reading ability, suggesting that social–emotional skills, as essential components of social–cognitive processes, could influence individuals' reading ability. In turn, this developed reading ability could influence the development of children's social–emotional skills during early and middle childhood [2]. We also examined the moderation effects of types of social–emotional skills, types of reading ability, and grade level on the correlation between social–emotional skills and reading ability. Results indicated that, compared to word reading, reading comprehension had a stronger correlation with social–emotional skills, which supports the simple view of reading (SVR) and suggests that reading comprehension is more complicated than word reading [9,129]. Future studies should explain the difference in the correlation between social–emotional skills and reading comprehension and between social–emotional skills and word reading.

Furthermore, our findings have practical implications for social–emotional intervention programs to improve children's reading ability. One previous study indicated that social–emotional intervention for preschool students from low SES families contributed to their reading development in grade one, even after controlling for vocabulary and emergent literacy skills [130]. Findings in previous studies, combined with those in this meta-analysis, suggest that social–emotional skills play important roles in the reading ability development of students in various grade levels, ranging from kindergarten to university and above. Parents and teachers should create opportunities that explicitly enhance social–emotional skills and integrate the growth of these skills with the development of students' reading ability at home or in the classroom. However, it should be noted that the correlation between social–emotional skills and reading ability was moderate, ranging from 0.20 to 0.29. Translating the correlations into variances, social-emotional skills accounted for a small percentage of variance in reading ability, ranging from 4.00% to 8.41%. These numbers indicate that although a social–emotional intervention program can contribute to reading development, the effects may be small, suggesting that social–emotional intervention alone may not be sufficient to improve reading ability [2]. The best approach to improving reading ability may be to combine social–emotional and reading intervention.

Our findings also have implications for reading intervention programs to improve children's social–emotional skills. Specifically, since readers need to comprehend the emotions of the characters and the relationships among these characters, social–emotional skills showed a stronger correlation with reading comprehension than with word reading [119].

Thus, compared to word reading, reading comprehension intervention may have a stronger effect on social–emotional skills.

In summary, this meta-analysis examined the correlation between social–emotional skills and reading ability and the potential moderators of this relationship. The results indicated that social–emotional skills were moderately correlated with reading ability and that social–emotional skills had a stronger correlation to reading comprehension than word reading. These findings suggest that teachers, educators, and parents should consider the roles of social–emotional skills in children's reading development and, conversely, the role of reading ability in developing one's social–emotional skills. High social–emotional skill levels could improve children's reading ability and general wellbeing.

**Author Contributions:** Conceptualization, X.T.; methodology, J.J.Y. and L.Y.; software, L.Y.; validation, L.Y. and J.J.Y.; formal analysis, J.J.Y. and L.Y.; investigation, L.Y.; resources, L.Y.; data curation, L.Y.; writing—original draft preparation, X.T. and L.Y.; writing—review and editing X.T. and L.Y.; visualization, X.T. and L.Y.; supervision, X.T. and J.J.Y.; project administration, X.T. and L.Y.; funding acquisition, X.T. All authors have read and agreed to the published version of the manuscript.

**Funding:** This research was supported by Seed Fund Grant, the Education University of Hong Kong with grant number RG60/21-22R.

**Data Availability Statement:** Data for performing analyses are available; please email us (xhtong@eduhk.hk) for access to the data.

**Conflicts of Interest:** The authors declare no conflict of interest.

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
