# Peer review of "Social–Emotional Skills Correlate with Reading Ability among Typically Developing Readers: A Meta-Analysis"

_education, doi:10.3390/educsci13020220_

Round 1

Reviewer 1 Report

I know is a lot of work behind this study and a lot of research hours to dig in www. 

Author Response

1. I know is a lot of work behind this study and a lot of research hours to dig in www. 

AUTHORS’ RESPONSE: Thanks very much for the reviewer’s positive and encouraging comments on our study.

Reviewer 2 Report

It is suggested that the authors include recommended practical application of their findings i.e. should socio-emotional skills be improved to improve reading comprehension and for what grade / age level etc.

Author Response

1. It is suggested that the authors include recommended practical application of their findings i.e. should socio-emotional skills be improved to improve reading comprehension and for what grade / age level etc.

AUTHORS’ RESPONSE: Thank you very much for the reviewer’s helpful suggestions. As you suggested, we have added some sentences on the practical application of our findings in the revised manuscript (see p. 13).

Reviewer 3 Report

Remove vague statement like Despite these limitations, this meta-analysis was the first to systematically examine 

Remove definition like Publication bias means that studies with statistically significant results are preferred 420 by journals for publication (Sterne et al., 2001), as it sounds vague

Discussion section must be improved

Correct grammar mistakes 

Accepted with minor changes

Author Response

1. Remove vague statement like Despite these limitations, this meta-analysis was the first to systematically examine 

Remove definition like Publication bias means that studies with statistically significant results are preferred 420 by journals for publication (Sterne et al., 2001), as it sounds vague

AUTHORS’ RESPONSE: Thank you very much for the helpful suggestion. We have modified these sentences in the revised manuscript (see p.9 and p13).

2. Discussion section must be improved

AUTHORS’ RESPONSE: Thank you very much for the reviewer’s helpful suggestions. We have revised the discussion part in the revised manuscript (see p.10-13).

3. Correct grammar mistakes 

AUTHORS’ RESPONSE: Thank you very much for the helpful suggestion. We have invited a native English speaker to help proofread the whole paper.